# GADD34 Ablation Exacerbates Retinal Degeneration in P23H RHO Mice

**DOI:** 10.3390/ijms232213748

**Published:** 2022-11-09

**Authors:** Irina V. Saltykova, Assylbek Zhylkibayev, Oleg S. Gorbatyuk, Marina S. Gorbatyuk

**Affiliations:** Department of Optometry and Vision Science, School of Optometry, University of Alabama at Birmingham, Birmingham, AL 35294, USA

**Keywords:** retinal degeneration, unfolded protein response, integrated stress response, p-eIF2α, translational attenuation, GADD34

## Abstract

The UPR is sustainably activated in degenerating retinas, leading to translational inhibition via p-eIF2α. Recent findings have demonstrated that ablation of growth arrest and DNA damage-inducible protein 34 (GADD34), a protein phosphatase 1 regulatory subunit permitting translational machinery operation through p-eIF2α elevation, does not impact the rate of translation in fast-degenerating *rd*16 mice. The current study aimed to validate whether P23H RHO mice degenerating at a slower pace manifest translational attenuation and whether GADD34 ablation impacts the rate of retinal degeneration via further suppression of retinal protein synthesis and apoptotic cell death. For this study, mice were examined with ERG and histological analyses. The molecular assessment was conducted in the naïve and LPS-challenged mice using Western blot and qRT-PCR analyses. Thus, this study demonstrates that the P23H RHO retinas manifest translational attenuation. However, GADD34 ablation resulted in a more prominent p-eIF2a increase without impacting the translation rate. GADD34 deficiency also led to a reduction in scotopic ERG amplitudes and an increased number of TUNEL-positive cells. Molecular analysis revealed that GADD34 deficiency reduces the expression of p-STAT3 and *Il-6* while increasing the expression of *Tnfa*. Overall, the data indicate that GADD34 plays a multifunctional role. Under chronic UPR activation, GADD34 acts as a feedback player, dephosphorylating p-eIF2a, although this role does not seem to be critical. Additionally, GADD34 controls cytokine expression and STAT3 activation. Perhaps these molecular events are particularly important in controlling the pace of retinal degeneration.

## 1. Introduction

Retinal degenerative diseases comprise a wide heterogeneous group of retinal degenerative disorders, including inherited retinal degeneration (RD), affecting more than 200,000 Americans and individuals worldwide [1]. Retinitis pigmentosa (RP) is a group of rare eye dystrophies that, when inherited, lead to a gradual loss of rods and cones in the retina. Over 3000 mutations in more than 50 different genes result in a nonsyndromic form of RP, while over 1200 other pathogenic mutations cause syndromic forms, for example Usher and Bardet–Biedl syndromes [1]. Primarily, RP affects rod photoreceptors, leading to retinal dysfunction and cell death and causing the loss of night vision, followed by cone photoreceptor deterioration.

Mutations in rod photoreceptor genes encode aberrant proteins that lack the ability to properly fold. These synthesized misfolded proteins manifest compromised protein trafficking to cellular compartment destinations, such as the outer segments (OSs) of photoreceptors. Unable to transport to the OS and consistently misfolded, proteins accumulate in the endoplasmic reticulum (ER), thus compromising this organelle’s homeostasis and activating a cellular program called the unfolded protein response (UPR). Mutant rhodopsin (RHO) is one of these proteins. The stress caused by misfolded proteins activates the protein kinase RNA-like endoplasmic reticulum kinase (UPR PERK) arm, promoting the phosphorylation (p) of eukaryotic initiation factor 2 alpha (eIF2α). In turn, p-eIF2α promotes a proadaptive signaling pathway—a temporal attenuation of global protein synthesis—whose major function is to cope with stress through adjustment of the translational rate. Overall, the PERK→p-eIF2α arm is also known as an integrative stress response (ISR).

Because retinal degenerative diseases are characterized by the consistent production of misfolded proteins, sustained ISR activation places a long-lasting hold on the recovery of protein synthesis in the deterioration of photoreceptor cells, leading to chronic translational attenuation. Recently, we investigated *rd16* mice deficient in growth arrest and DNA damage-inducible Protein 34 (GADD34^−/−^). These mice expressed truncated CEP290, manifested chronic ISR activation, and developed severe RD rapidly, thus modeling Leber congenital amaurosis 10 and RP in humans. We demonstrated that GADD34-mediated manipulation with the p-eIF2α level in *rd16* photoreceptors does not modify the translation rate and has no major impact on retinal function in mice with ciliopathies, although GADD34 ablation slowed down the rate of apoptotic cell death [2]. This is why we have become interested in understanding whether the GADD34-mediated elevation of p-eIF2α in retinas degenerating at a slower pace alters the progression of retinal dystrophy in mice.

Among several animal models of RP, there are P23H RHO mice with a relatively later onset and slower rate of RD. The proline to histidine at position 23 (P23H) RHO mutation accounts for about 12% of all autosomal dominant (ad) RP in North America [3,4]. It has been reported that human P23H RHO knock-in mice manifest accelerated photoreceptor cell degeneration and display decreased lengths of their OS [5]. In addition, the P23H protein is inadequately glycosylated with about 10% of WT RHO, suggesting that the mutation affects the consensus sequence for glycosylation at Asn^15^. Progressive retinal degeneration in these mice is obvious at Postnatal Day (P) 35. At this time point, a few rows of photoreceptor nuclei are missing and the amplitudes of the electroretinogram (ERG) recording (retinal function test) are significantly diminished as compared to the wild-type retina. These mice recapitulated the human retinopathy and showed rod photoreceptor functional deficit as compared to the cone system [5]. The protein structural study confirmed inefficiency in the signaling in P23H RHO, a perturbance in the P23H RHO protein dimerization, and a drastically altered mechanism of activation [6]. This pointed out that the ruptured Cys110–Cys187 disulfide bond in the extracellular loop 2 regions, the retinal binding site, of the protein structure is the source of extreme misfolding of P23H RHO through the entire receptor [6]. The P23H RHO mice demonstrated both activated ISR and an elevated p-eIF2α level [7].

The GADD34 protein has a multitude of other roles besides regulation of eIF2α activity including control of apoptosis. Moreover, both the inhibition and activation of apoptosis by GADD34 have been reported [8,9,10]. In addition to apoptosis, GADD34 is proposed to regulate STAT3. Thus, the study by Tanaka et al. proposed the GADD34 protein to regulate inflammation and host defense systems through the IL-6/STAT3 axis [11]. Moreover, both visual dysfunction and decreased RHO levels have been shown to correlate with increased STAT3 activation [12]. Therefore, in this study, we generated P23H RHO GADD3434^−/−^ mice lacking the PPP1R15A regulatory subunit of phosphatase holoenzyme protein phosphatase 1 (PP1)/GADD34 that dephosphorylates p-eIF2α to study its role in the RD of P23H RHO mice. This action was expected to elevate the p-eIF2α level in degenerating photoreceptors and to modulate the translation rate. We hypothesized that, similar to *rd16* mice, the P23H RHO mice degenerating at a slower pace manifest the suppression of the translational rate, and the GADD34-mediated p-eIF2α elevation does not impact the rate of protein synthesis in their retinas; however, unlike *rd16* mice, P23H RHO mice may require GADD34 to control apoptotic cell death. The data of this study indicate that the elevation of p-eIF2α in the P23H RHO GADD34^−/−^ retina does not lead to a more pronounced inhibition of protein synthesis, thus confirming the results obtained with *rd*16 mice and pointing out that, during chronically activated UPR, perhaps a mode other than the PERK→p-eIF2α mode controls the translational rate in degenerating photoreceptors [2,13]. In addition, the study revealed the anti-apoptotic role of GADD34 in P23H RHO GADD34^−/−^ photoreceptors with ongoing ISR.

## 2. Results

### 2.1. GADD34 Controls the Rate of RD in P23H RHO Retinas, Resulting in More Severe Retinal Degeneration

The p-PERK→p-eIF2α arm of UPR signaling is chronically upregulated in P23H RHO retinas starting from P19 [7]. Therefore, the p-eIF2α level was manipulated by ablating the PP1 regulatory subunit GADD34. Thus, it was found that a deficiency in GADD34 in the retina accelerated RD in P23H RHO mice. The a- and b-wave amplitudes of the scotopic ERG were registered, and it was found that GADD34 ablation reduced the a-wave amplitude as compared to the control, degenerating the retina by 28% (Figure 1A,B, *p* < 0.05, *n* (number of animals) = 4–5), while no difference in the b-wave recording was observed between the experimental and control groups. Given the GADD34 ablation results in a decrease in the number of apoptotic cells in the rd16 retina, we then performed the detection of apoptosis by the TUNEL assay in P23H RHO GADD34^−/−^ retinas. Surprisingly, we learned that the decline in the photoreceptor-originated ERG waveform was in agreement with an increase in the number of apoptotic cell deaths in the P22 experimental degenerating retinas, suggesting that, most likely, GADD34 contributes to photoreceptor cell viability (Figure 1C,D). Therefore, in the next step, the levels of chronically activated eIF2a and protein synthesis were identified in the P23H RHO retina.

### 2.2. GADD34 Ablation in P23H RHO Retinas Leads to an Increase in p-eIF2α and No Difference in the Rate of Protein Synthesis

First, it was found that GADD34 ablation results in dramatic upregulation of p-eIF2α in P23H RHO retinas by about twofold as compared to P23H RHO degenerating and wild-type retinas, respectively (Figure 2A, *p* < 0.0001, *n* = 4). Therefore, in the next step, the translational rate in P23H RHO GADD34^−/−^ retinas was detected. First, it was observed that P23H RHO retinas with elevated p-eIF2α manifest a reduction in the rate of global protein synthesis (Figure 2B,C, *p* < 0.0001, *n* = 4). No difference was found in the density of incorporated puromycin between P23H RHO and P23H RHO GADD34^−/−^ retinas, suggesting that, despite the dramatic increase in p-eIF2α, the level of protein synthesis is not inhibited further in P23H RHO GADD34^−/−^ mice. In agreement with these findings, the level of RHO, a major photoreceptor-specific protein, was found to have declined in both degenerating retinas (Figure 2D, *p* < 0.0001 for P23H RHO, *p* < 0.0001 for P23H RHO GADD34^−/−^, *n* = 4). Given an increase in the terminal deoxynucleotidyl transferase dUTP nick end labeling (TUNEL)-positive photoreceptor cells and a lack of changes in the translational rate and the RHO level, how the retinal cells increasingly die in the P23H RHO GADD34^−/−^ retinas remained unclear.

### 2.3. GADD34 Controls STAT3 Activation and Cytokine Expression in P23H RHO Degenerating Retinas

It is known that both visual dysfunction and decreased RHO levels have been shown to correlate with increased STAT3 activation [12]. Given a decline in RHO levels in the P23H RHO retinas (Figure 2B), we became interested in the activation of the STAT3 transcriptional factor.

A significant over twofold elevation of p-STAT3 was observed in naïve P23H RHO retinas at P30 (Figure 3A, *p* < 0.0001, *n* = 4) as compared to C57BL6 retinas, which correlated with the decrease in the RHO level depicted in Figure 2D. Interestingly, in P23H RHO GADD34^−/−^ mice, the p-STAT3 level was 18% lower compared to the P23H RHO control (Figure 3A, *p* < 0.05, *n* = 4). These findings fueled our interest in whether the alteration in the p-STAT3 level correlates with cytokine expression in both RD mouse retinas. To compare the ability of both degenerating retinas to control cytokine expression, an inflammatory response was induced in the retinas by injecting LPS. First, it was found that the LPS-treated P23H RHO GADD34^−/−^ retinas manifest a 24% decline in p-STAT3 protein as compared to treated P23H RHO (Figure 3B, *p* < 0.001, *n* = 5), and this fact was consistent with p-STAT3 activation in naïve retina. Second, an over threefold decrease in *Il-6* and about a threefold increase in *Tnfa* mRNA expression in the LPS-treated P23H RHO retina-deficient GADD34 were found (Figure 3C, *p* < 0.05 for both groups, *n* = 5), suggesting that cytokine expression is regulated by GADD34.

Given that P23H RHO GADD34^−/−^ retinas manifest an increase in the number of TUNEL-positive cells, which has been proven to be evidence of the apoptotic nature of cell death, ionized calcium-binding adaptor-molecule-1 (Iba1)-positive microglia and macrophages accumulated in the LPS-treated retinas were analyzed. Surprisingly, no difference was observed in the Iba1-positive cells when the two degenerating retinas were compared at 24 h (Figure 3D,E), suggesting that GADD34 could regulate local cytokine expression.

Overall, the study demonstrated that GADD34’s operation in degenerating retinas with ongoing UPR activation is necessary to control the progression of retinal pathogenesis in P23H RHO mice; GADD34 ablation in P23H RHO retinas results in a decrease in retinal function and an increase in apoptotic cell death.

## 3. Discussion

ISR is chronically activated in the P23H RHO retinas. Starting at P19, the p-eIF2α level was persistently upregulated [7]. These findings suggest that the rate of translation could be chronically attenuated in these retinas. Indeed, the rate of protein synthesis was found to be diminished in the P23H RHO retinas at P30. These results supported our previous findings with rapidly degenerating *rd*16 and *rd*10 retinas, both showing a decline in the rate of translation [14] and indicating that, overall, chronically upregulated ISR in the retina results in attenuation of global protein synthesis in varied animal models of RD. The goal of the current study was to further test the hypothesis of whether manipulation with the p-eIF2α level could modify the rate of RD in mice deteriorating at a slower pace compared to *rd*16 and whether these changes are associated with further suppression of protein synthesis and an increase in apoptotic cell death.

GADD34, a protein phosphatase 1 regulatory subunit, plays a pivotal role in the regulation of eIF2α activity. Therefore, ablation of this protein is anticipated to provide sustained activation of eIF2α because PP1 phosphatase is no longer active. The data from this study showed an increase in p-eIF2α. Therefore, based on these findings, one can expect to observe the repression of protein synthesis on a larger scale. In contrast, the data of this study show that the further p-eIF2α increase does not inhibit global protein synthesis, thus supporting our published study demonstrating that GADD34 loss increases p-eIF2α, but has no impact on translation rates in RD [2]. This also implies that the PERK→p-eIF2α axis either does not play a major role in the regulation of protein synthesis or reaches a threshold of translational suppression, upon which further alteration in the p-eIF2α level does not affect the translational rate. For example, in the *rd*16 retinas, it was discovered that the mTOR/p-4EBP1/2 axis is a major mode of translational control [2]; therefore, a similar mode of translational regulation could operate in the P23H RHO retinas as well. The current and published studies of our group also indicate that, most likely, the inhibition of protein synthesis is not a cell defense mechanism in the retina by which deteriorating photoreceptors survive, but are the consequences of RD progression [13].

In the *rd*16 retinas, GADD34 ablation reduced the number of TUNEL-positive cells and the level of gliosis, although it did not result in a significant difference in scotopic ERG amplitudes [2]. In contrast, the P23H RHO GADD34^−/−^ retinas manifested robust TUNEL-positive staining as compared to the P23H RHO retinas. The results of this study also demonstrated that GADD34 is necessary for degenerating P23H RHO photoreceptors. GADD34 loss results in a small, but significant decline in scotopic a-wave ERG amplitudes. The increase in the number of TUNEL-positive cells supports these findings and provides evidence of more pronounced cell death in the P23H RHO GADD34^−/−^ retinas.

The primary functional role of GADD34 in cells with ER stress is to operate as a negative feedback loop during UPR activation, and its function allows for translation recovery through eIF2α dephosphorylation. Therefore, given the fact that the alteration in the p-eIF2α level does not affect the translational rate, how GADD34 controls retinal generation in mice was further explored in the current study. In addition to the PP1 regulatory function, GADD34 was shown to be required for normal cytokine production, both in vitro and in vivo. The cytokine production of IL-6 affected by GADD34 deletion/inactivation has been shown in multiple studies [11,15,16]. In this study, a reduction and increase in *Il-6* and *Tnfa* mRNA expressions, respectively, were observed in degenerating retinas challenged with LPS.

Although the translational rate of the IL-6 cytokine in naïve P23H RHO was not accessed, it is possible that GADD34 could selectively control protein synthesis in the degenerating retinas. For example, Tanako et al. demonstrated a reduction of pro-inflammatory IL-6 cytokine production after LPS stimulation in GADD34 KO mice with colitis-associated cancer [11]. Additionally, an alternative mechanism—internal-ribosome-entry-site (IRES)-mediated translational control—could regulate IL-6 expression. Thus, it has been proposed that GADD34 promotes IRES activity to regulate protein expression in addition to the regulation of phosphorylation of eIF2a [17]. Interestingly, the regulatory IRES sequence has been reported for the IL-6 gene (hsa_ires_00025.1; http://reprod.njmu.edu.cn/cgi-bin/iresbase/index.php). All these studies indicate that IL-6 expression could be regulated independently of the p-eIF2α→NF-kB→IL-6 axis, and the lack of GADD34 in P23H RHO LPS-treated mice could be responsible for the reduction of IL-6 cytokine in their retinas. Given the limitation of the current study, which did not provide the regulatory details of GADD34-mediated alteration of IL-6 and other individual protein expression escaping global translational attenuation, future experiments should shed light on the relationship between GADD34 and IL-6 in degenerating retinas.

In addition to IL-6, GADD34 controls *Tnfa* expression in P23H RHO retinas. It is known that preferential translation varies between stress. Under conditions of LPS-induced stress in P23H RHO GADD34^−/−^ retinas with enhanced p-eIF2a levels, TNF expression could occur in both p-eIFα-dependent and -independent manners. While studies demonstrated that TNFα expression is induced by the p-eIF2α→NF-kB→TNFa axis, other reports indicated that induction of p-eIF2α is not involved in the control of TNFα expression [18,19]. Thus, the research conducted in epithelial cells demonstrated that the application of guanobenz, a selective PP1 inhibitor, results in TNFα inhibition, and the mechanism of such activation is associated with upstream molecular events, such as interfering with the transforming growth factor (TGF)-*β*-activated kinase 1 (TAK1) complex [18]. Another study suggested that the PP1 regulatory subunit GADD34 can directly dephosphorylate IKK (Ser180/Ser181), returning the kinase to its default, inactive stage, thus regulating the downstream cytokine network [20]. This study implies that, in addition to the p-eIF2α→NF-kB→TNFa axis, TNFa could be regulated independently of the p-eIF2 level through GADD34-mediated phosphorylation of IKK. Therefore, the observed TNFα induction in the LPS-treated P23H RHO GADD34^−/−^ retinas with an increase in p-eIF2α could be a result of dual events: the activation of the p-eIF2α→NF-kB→TNFa axis and an increase in p-IKK-induced TNFα expression as a result of PP1 deficiency. Nevertheless, the TNFα reduction is beneficial to cone survival in mice with RD [21]. Therefore, a future study should be conducted to investigate the control mode of TNFα expression in the P23H RHO retina.

GADD34 mediates *Il-6* expression. The IL-6 cytokine has been shown to regulate p-STAT3 [22,23]. The phosphorylation has been proposed to occur through JAK-mediated activation of STAT3 in response to stimulation by cytokines or growth factors [24]. We learned that p-STAT3 was also diminished in the LPS-treated retinas. Moreover, the p-STAT3 level declined in naïve P23H RHO retinas with GADD34 ablation. Altogether, these data demonstrate that GADD34 is upstream of STAT3 activation, which has been shown to act both as a regulator of retinal cell survival [25,26] and an inducer of retinal degenerative changes upon long-term activation [27].

Although the relationship between eIF2α and STAT3 has not been well investigated, it has been demonstrated that the activated STAT3 interacts with PERK and inhibits the phosphorylation of eIF2α [28]. In agreement with this study, we found a reduction of p-STAT3 in P23H RHO GADD34^−/−^ retinas with elevated p-eIF2α. Nevertheless, despite the observed reciprocal regulation of p-STAT3 and p-eIF2a, we believe that the major cause of p-eIF2α elevation is GADD34 deficiency. Therefore, to further assess the role of p-STAT3 in the RD of P23H RHO mice, one should trace the localization and phosphorylation patterns of the p-STAT3 protein in the course of RD.

Interestingly, the phosphorylation position seems to play an important role in the distribution of p-STAT3. Two major phosphorylation sites are known for STAT3: Y705 and S727. The phosphorylated STAT3 (S727) protein is primarily transported to mitochondria and is responsible for retinal ganglion cell (RGC) axon regeneration upon stress [29]; p-STAT3 (Y705) is localized in the nucleus and found in the retinas of animal models of RD. For example, a recent study conducted in the Gross lab showed Q344X RHO knock-in mouse retinas manifesting sustained p-STAT (Y705) upregulation during RD confined to the nuclei of Müller cells in the retinal inner nuclear layer (INL) [30]. Other retinal degenerative models, transgenic RHO P347S and Prp2rds^+/−^ (perpherin-2, retinal degeneration slow) mice, manifest similar p-STAT3 (Y705) levels in Müller glia, in addition to a weak immunoreactivity for p-STAT3 in photoreceptors [26]. In contrast, a group led by John Ash identified p-STAT3 (Y705) in photoreceptors treated with leukemia inhibitory factor [31]. It needs to be emphasized that both conducted studies are in agreement with the prosurvival role of STAT3 activation in degenerating retinas [26,31]. These studies, together with the previous findings, indicate that GADD34 may play a local role in degenerating retinas. As evidence, GADD34 regulates *Il-6* expression and p-STAT3 activation in the retina. Deficiency in GADD34 leads to a reduction in p-STAT3 and the accelerated rate of photoreceptor cell death in P23H RHO retinas.

In conclusion, it is hypothesized that GADD34 plays a multifunctional role in the degenerating retina. On the one hand, under chronic UPR activation, GADD34 acts as a feedback player by dephosphorylating p-eIF2α, although this role does not seem to be critical due to the lack of further translational repression in degenerating retinas. On the other hand, GADD34 controls the IL-6/STAT3 arm and STAT3 activation, a lack of which compromises photoreceptor cell viability, leading to a decrease in the scotopic a-wave amplitude and an increase in apoptotic cell death in degenerating retinas. These late events may control the pace of RD in the P23H RHO retinas.

## 4. Materials and Methods

### 4.1. Animals

The P23H RHO and C57BL6 mice were purchased from the Jackson Laboratory (Bar Harbor, ME, USA). The GADD34^−/−^ mice were produced as previously described [2]. Then, the P23H RHO mice were crossed with GADD34^−/−^ and housed in the UAB animal core facility with a 12 h light/dark cycle with free access to a standard diet and water. Retinas from control and experimental mice of both sexes were used at Postnatal Days (P) 22 and 30. Mice were euthanized by CO_2_ asphyxiation followed by cervical dislocation. All animal experiments followed a protocol (IACUC#21044) approved by the University of Alabama at Birmingham Institutional Animal Care and Use Committee (IACUC) and conformed to guidelines from the Association of Research in Vision Science and Ophthalmology.

### 4.2. Electroretinography 

To conduct the scotopic electroretinography (ERG) recording, the LKC BigShot ERG (Gaithersburg, MD, USA) instrument was used. To that end, the mice were dark-adapted overnight and then anesthetized with an intraperitoneal injection of ketamine/xylazine on the basis of their weight. Topical 2.5% phenylephrine (Paragon BioTeck, Inc., 42702–102-15, Portland, OR, USA) and Gonak, 2.5% sterile hypromellose ophthalmic demulcent solution (AKORN, Lake Forest, IL, USA), were applied to the corneal surfaces of the mice. Then, we placed a monopolar contact loop on the surface of the cornea to register the retinal a- and b-wave ERG amplitudes. The reference electrodes were placed under the scalp and in the tail. Then, the mice were exposed to five flashes of 0.025 cd*s/m^2^, 2.5 cd*s/m^2^, 7.91 cd*s/m^2^, and 25 cd*s/m^2^ light intensities with intervals of 45 s. Their ERGs were recorded simultaneously. The a- and b-waveforms were analyzed using the LKC EM software (Gaithersburg, MD, USA).

### 4.3. Histology and Terminal Deoxynucleotidyl Transferase dUTP nick-End Labeling Assay

The eyes of the mice were enucleated at P22 and placed in 4% paraformaldehyde (PFA) (Santa Cruz Biotechnology Inc., sc-281692, Dallas, TX, USA). A 33G needle was used to create a small hole for eye tissue fixation in 4% PFA for 1 h. Then, we washed the eyes with phosphate buffered saline (PBS) (Gibco, 10010-023, Paisley, UK) and immersed them in 30% sucrose overnight. The eyes were further examined for cryopreservation in an optimal cutting temperature (OCT) compound (VWR: 25608–930). The 12 μM retinal cryosections were performed using a sectioning system (Leica CM 1510S; Leica, Buffalo Grove, IL, USA).

To detect apoptotic cell death, TUNEL staining (Click-iT Plus TUNEL assay, ThermoFisher Scientific, C10617, Waltham, MA, USA) was performed according to the protocol of the manufacturer’s procedure on retinal sections from mice at P22. Whole eyes were removed and fixed in 4% PFA for 20 min on ice, and a needle (33G) was inserted at the limbus to create a small hole, following incubation for 2 h. Then, the fixed eyes were placed in 30% sucrose in PBS overnight at 4 °C. Eyes were cryopreserved in Tissue-Tek OCT compound (Sakura Fintek USA Inc., 4583, Torrance, CA, USA) and kept at −80 °C until sectioning. The samples were cut into 12 μm-thick sections on a CM 1510S cryostat (Leica, Deer Park, IL, USA).

### 4.4. LPS Experiment

The P30 mice were injected with 10 mL/kg (stock solution of 1 mg/mL) of lipopolysaccharide (LPS) intraperitoneally at P30, as previously described [32]. After 6 and 24 h, the mice were sacrificed, and their eyeballs were enucleated for Western blot and qRT-PCR analyses, respectively. At 24 h post-injection, immunostaining to identify Iba1-positive cells was also performed. For Iba1-positive microglial cell counting, the cryosections were rinsed with PBS and then blocked in 5% normal donkey serum and 0.3% Triton X-100 in PBS for 1 h at room temperature. The retinal cryosections were then incubated in the primary anti-IBA-1 Antibody (Wako Chemicals, 019-19741, Osaka, Japan). The secondary Alexa Fluor 555 or 488 anti-rabbit antibodies (ThermoFisher Scientific, A31572; A21206, Eugene, OR, USA) diluted in PBS (1:500) were applied for 2 h at room temperature. Nuclei were visualized by counterstaining with DAPI in a fluorescence mounting medium (Southern Biotech, 0100-20, Birmingham, AL, USA), and images were acquired with a Keyence BZ-X800 Fluorescence Microscope (Itasca, IL, USA). Investigators blinded to the results counted the number of Iba1-positive cells in the retinal sections. Fluorescent images were taken, and the Iba1-positive cells were calculated in a vision field of 200 µm^2^.

### 4.5. Analysis of Nascent Protein Synthesis

The SUrface SEnsing of Translation (SUnSET) method has been described previously [33]. Briefly, the mice were injected with puromycin (puromycin dihydrochloride; Santa Cruz Biotechnology, CAS 58–58–2, Dallas, TX, USA) intraperitoneally at a dosage of 0.04 μmol/g body mass. Then, mice were sacrificed, and their retinas were harvested to prepare protein extracts with RIPA buffer (Cell Signaling, 98065, Danvers, MA, USA). Both retinas from each mice were combined into one sample. Total protein (40 μg per well) was electrophoresed on a 4–12% SDS-polyacrylamide gradient gel (Bio-Rad, Hercules, CA, USA) and blotted onto a PVDF membrane. The antibodies specific to puromycin (mouse, MABE343, Millipore Sigma, St. Luis, MO, USA) were applied to incubate the membranes. A secondary antibody specific to IgG2a (goat antimouse peroxidase affinipure IgG, Fcγ Subclass 2a Specific: 115-035-206, Jackson Immuno Research Laboratories Inc.; West Grove, PA, USA) was reused to detect the bands. After scanning the density of the puromycin-incorporated bands using a LI-COR Odyssey XF imager (Lincoln, NE, USA), the membranes were stained with Coomassie Brilliant Blue R-250 Stain (Bio-Rad, 1610436, Hercules, CA, USA) for their normalization. The relative densities of the entire lanes of Coomassie-stained proteins and of incorporated puromycin were calculated using the ImageJ software and used to quantify the rate of protein synthesis.

### 4.6. Immunoblotting

The mouse retinas were dissected and lysed with RIPA buffer supplemented with a 1% Halt Protease Inhibitor and a phosphatase inhibitor cocktail (Thermo Fisher Scientific, 87786, Waltham, MA, USA). The study then proceeded with the Western blot techniques described in [33]. Protein samples (40–60 μg) were separated by SDS-PAGE and electroblotted to a PVDF membrane. The following antibodies were used: rabbit phospho-eIF2α (p-S51, 3398), eIF2α (9722), rabbit phospho-STAT3 (Y705, 9131S), rabbit STAT3 (12640S) from Cell Signaling Technology (Danvers, MA, USA), mouse Anti-RHO (1D4) invented at the University of British Columbia (UK), rabbit anti-Actin (A2066), mouse anti-β-Actin (#A2228) from Sigma-Aldrich (St. Luis, MO, USA). To detect p-STAT3 in LPS-injected mice, LPS injections and the preparation of retinal extracts were conducted as previously described [32].

### 4.7. Real-Time Quantitative PCR

The retinas were dissected and frozen in liquid nitrogen. Samples were stored at −80°C until processing. RNA was isolated from the retinas using TRIzol (ThermoFisher Scientific, 15596026, Waltham, MA, USA). Genomic DNA was removed using ezDNAse cDNA was synthesized from 500 µg of RNA using SuperScript IV VILO Master Mix (ThermoFisher Scientific, 11766050, Waltham, MA, USA). A cDNA synthesis reaction mixture containing no reverse transcriptase was used as a negative control in subsequent amplifications to confirm the absence of genomic DNA contamination. The following TaqMan primers (ThermoFisher Scientific, Waltham, MA, USA) were used: *Il6* (Mm00446190_m1), *Tnfα* (Mm99999068_m1), *GAPDH* (Mm99999915_g1). Quantitative RT-qPCR was performed in duplicate using iTaq Universal Probe Supermix (Bio-Rad, 1725131, Hercules, CA, USA) on a QuantStudio 3, 96 well plate Real time PCR machine (ThermoFisher Scientific, A28131, Waltham, MA, USA). Gene expression was calculated using the 2^−∆∆CT^ method and normalized against the expression of *GAPDH*.

### 4.8. Statistics

The Student *t*-test was used to compare two groups, and ordinary one-way ANOVA (Tukey’s test) and two-way ANOVA (Sidak’s test) with multiple comparisons were carried out to compare three groups of animals. All statistics were calculated using GraphPad Prism 9 software (San Diego, CA, USA).

## 5. Conclusions

In conclusion, it was hypothesized that GADD34 ablation exacerbates retinal degeneration in P23H RHO mice. We found that this protein plays a multifunctional role in degenerating retina. On the one hand, under chronic UPR activation, GADD34 acts as a feedback player by dephosphorylating p-eIF2α, although this role does not seem to be critical due to the lack of further translational repression in degenerating retinas. On the other hand, GADD34 controls the IL6/STAT3 arm and STAT3 activation, a lack of which compromises photoreceptor cell viability, leading to a decrease in the scotopic a-wave amplitude and an increase in apoptotic cell death in degenerating retinas. These late events may control the pace of RD and apoptotic cell death in the P23H RHO retinas.

## Figures and Tables

**Figure 1 ijms-23-13748-f001:**
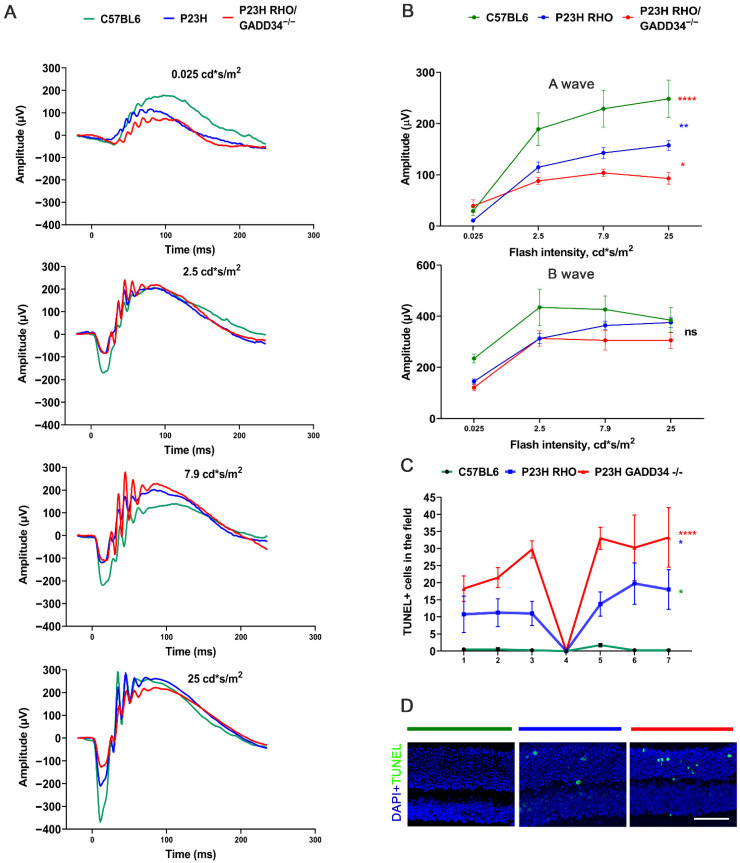
GADD34 deficiency results in a faster degenerating pace in P23H RHO retinas at P30. (**A**) The a-wave of the scotopic ERG amplitude was diminished in the P23H RHO GADD34^−/−^ retinas (*n* = 5). (**B**) The representative a- and b-waves’ ERG recording at 0.025; 2.5; 7.9; 25 cd*s/m^2^ light intensity. (**C**) The impairment of retinal function is accompanied by a significant increase in the number of TUNEL-positive nuclei in P23H RHO GADD34^−/−^ mice compared to P23H RHO and C57BL6 retinas at P22. The number of TUNEL-positive nuclei in the vision field (200 µm^2^) is shown (*n* = 4). (**D**) Images of the retinas of C57BL6, P23H RHO, and P23H RHO GADD34^−/−^ mice depicting TUNEL-positive nuclei. * *p* < 0.05, ** *p* < 0.01, and ****** *p* < 0.0001, “ns”—not significant. The scale bar is 50 µm.

**Figure 2 ijms-23-13748-f002:**
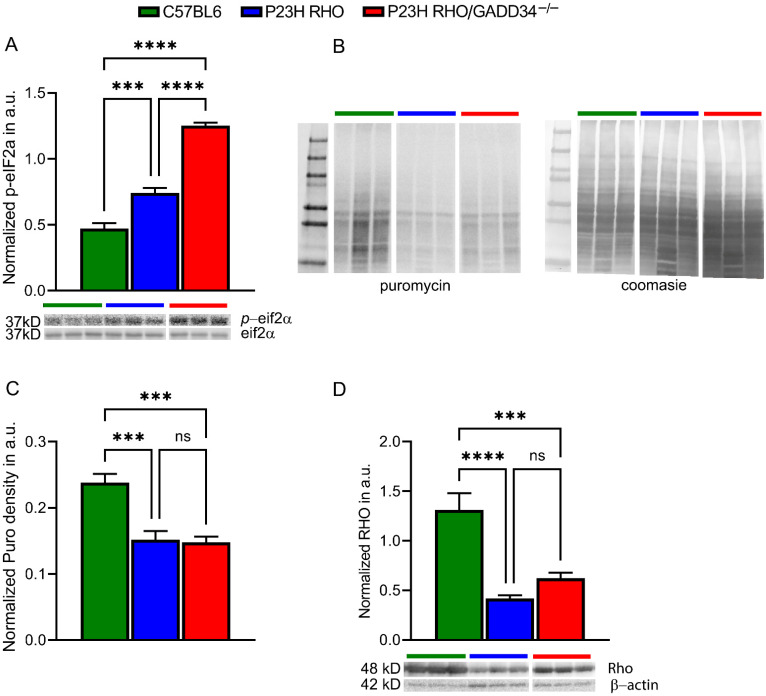
GADD34 deficiency results in an increase in p-eIF2α levels and no further suppression of the translational rate in P23H RHO retinas. (**A**) Both the P23H RHO and P23H RHO GADD34^−/−^ retinas demonstrated an increase in p-eIF2a (*n* = 4). Ablation of GADD34 in the P23H RHO retinas resulted in about a twofold elevation compared to the P23H RHO and C57BL5 control groups, respectively. (**B**) Using the SUnSET method with puromycin incorporated into nascent proteins in the retinas, newly synthesized proteins (*n* = 4) were measured. Images of the Western blot stained with anti-Puro antibody and Coomassie were used to normalize the density of the incorporated puromycin. Puromycin immunoblotting was normalized to Coomassie-stained proteins. (**C**) A graph comparing the levels of translation in the C57BL6, P23H RHO, and P23H RHO GADD34^−/−^ retinas is shown. It was found that P23H RHO retinas manifest translational attenuation at P30 as compared to C57BL6 retinas. GADD34 ablation does not result in further translational repression in P23H RHO mice, although the p-eIF2α is dramatically elevated. (**D**) Measurement of RHO protein in the wild-type and degenerating retinas revealed a significant decline in RHO levels in both the P23H RHO and P23H RHO GADD34^−/−^ mice (*n* = 4). Relative density measurements correspond to the intensities of the immunoblotting bands or lanes normalized to the β-actin control. Data are shown as the mean ± SEM. a.u.: arbitrary units, *** *p* < 0.001, ****** *p* < 0.0001 and “ns”—not significant.

**Figure 3 ijms-23-13748-f003:**
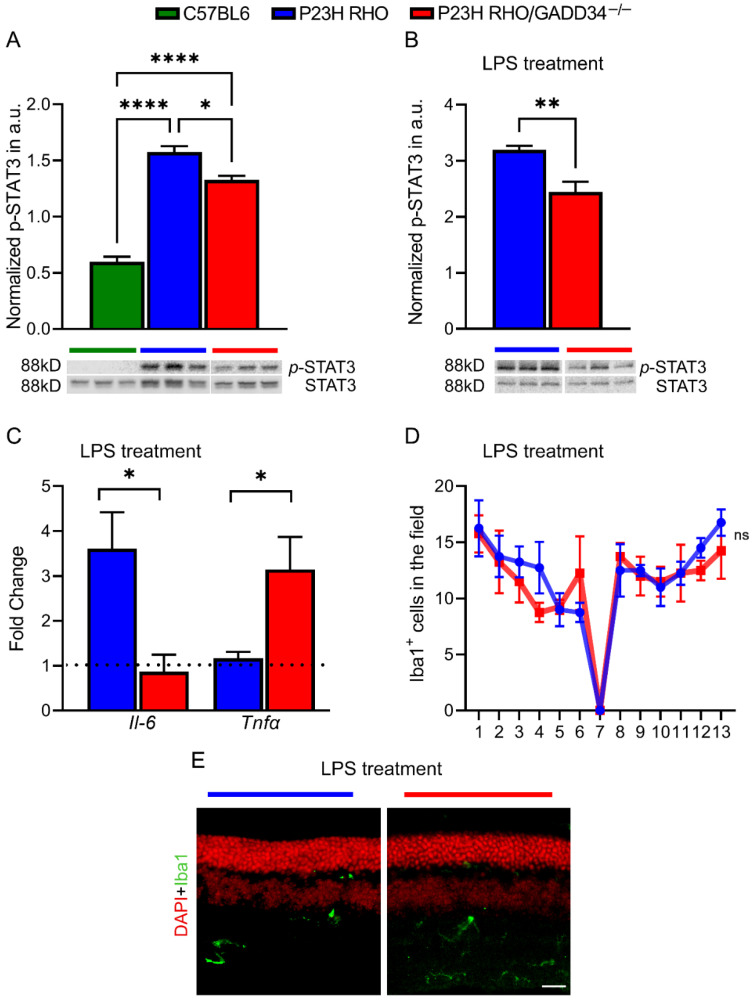
GADD34 controls STAT3 activation and cytokine expression in P23H RHO retinas. (**A**) The level of activated STAT3 (Y705) was significantly elevated in naïve P23H RHO retinas at P30 compared to C57BL6 mice with GADD34 ablation in these animals, resulting in a decline in the level of p-STAT3 (*n* = 4). Images of Western blots probed with anti-p-STAT3 and STAT3 antibodies are shown. Relative density measurements correspond to the intensities of the immunoblotting bands normalized to the STAT3 control. (**B**) The treatment with LPS revealed a difference in the p-STAT3 level between the two groups of degenerating mice in 6 h. Western blot images are also shown. (**C**) The decline in p-STAT3 in the P23H RHO GADD34^−/−^ retinas was accompanied by a decrease in *Il-6* and an increase in *Tnfa* expression in 24 h with LPS treatment (*n* = 5). (**D**) Interestingly, despite the cytokine and STAT3 regulation by GADD34, the number of macrophages and microglia in the retinas of the LPS-administered mouse groups did not change. (**E**) Images of LPS-treated mice retinas stained with anti-Iba1 antibody. The scale bar is 20 µm. * *p* < 0.05, ** *p* < 0.01, and ****** *p* < 0.0001, “ns”—not significant.

## Data Availability

Data are available upon request from the authors.

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
