# Peer review of "GADD34 Ablation Exacerbates Retinal Degeneration in P23H RHO Mice"

_ijms, 2022, doi:10.3390/ijms232213748_

Round 1
Reviewer 1 Report (New Reviewer)
The manuscript entitled « Elevation of p-eIF2a Does Not Control the Translation Rate in Degenerating Retinas with an Ongoing Integrated Stress Response », by Saltykova and colleagues, examines whether manipulating translational rates through modulating regulatory pathways has an effect on retinal degeneration rates. Overall the paper is well written and illustrated, albeit providing negative results. My comments are given below.
Some minor english correction is necessary, eg. « Therefore, the study was proceeded by generating the P23H RHO GADD34−/−… » (l. 89, p. 2).
This reviewer is a bit confused about the hypothesis : early on the authors say the aim of the study was to see whether affecting translational rate could slow down degeneration in mouse models with slow degeneration (since it doesn’t have any impact in rapid degenerations) ; but then they say « Therefore, we hypothesized that similar to rd16 mice, the P23H RHO mice degenerating at a slower pace manifest suppression of translational rate and further GADD34-mediated p-eiF2α elevation does not impact the rate of protein synthesis in their retinas. » These two statements seem to be in contradiction ? Furthermore, the study seems to stray away from this initial idea (presumably because the results were negative), going into inflammatory pathway involvement. And finally the authors suggest a dual role for GADD34, the first one non-essential and the second in regulation of cytokine responses. So although the data are clear and convincing, the aims are a bit rambling and should be better explained, maybe using sub-headings for each aspect.
Author Response
Dear Reviewers,
Thank you very much for your time spent reading the manuscript and writing the comments. We also appreciate your concerns. We agree with all the comments and modified the text. We hope that the revision makes the manuscript stronger and of better quality.
Reviewer #1:
The manuscript entitled « Elevation of p-eIF2a Does Not Control the Translation Rate in Degenerating Retinas with an Ongoing Integrated Stress Response », by Saltykova and colleagues, examines whether manipulating translational rates through modulating regulatory pathways has an effect on retinal degeneration rates. Overall the paper is well written and illustrated, albeit providing negative results. My comments are given below.
Our response: Thank you.
Q1: Some minor english correction is necessary, eg. « Therefore, the study was proceeded by generating the P23H RHO GADD34−/−… » (l. 89, p. 2).
Our response: Thank you. We edited.
Q2: This reviewer is a bit confused about the hypothesis: early on the authors say the aim of the study was to see whether affecting translational rate could slow down degeneration in mouse models with slow degeneration (since it doesn’t have any impact in rapid egenerations) ; but then they say « Therefore, we hypothesized that similar to rd16 mice, the P23H RHO mice degenerating at a slower pace manifest suppression of translational rate and furtherGADD34-mediated p-eiF2α elevation does not impact the rate of protein synthesis in their retinas. » These two statements seem to be in contradiction. Furthermore, the study seems to stray away from this initial idea (presumably because the results were negative), going into inflammatory pathway involvement. And finally the authors suggest a dual role for GADD34, the first one non-essential and the second in regulation of cytokine responses. So although the data are clear and convincing, the aims are a bit rambling and should be better explained, maybe using sub-headings for each aspect.
Our response: Thank you. We restated the aim and hypothesis, and they are now consistent throughout the manuscript.
Reviewer 2 Report (New Reviewer)
In this manuscript, the authors explored the role and potential mechanism of GADD34 in the cell apoptosis of P24H retinas. However, there are several questions still unclear and the authors should give potential explanations.
1. In figure 1A and B, the ERG analysis only showed the results for the scotopic a/b wave at the light strength of 25. However, in the method part, the ERG study of retinal responses was done at several light strengths. How about those results?
2. In figure 2A, there was a significant increase of eif2a and p-eif2a in GADD34-/- retina compared with the P23H RHO retina, however, the authors didn’t provide potential explanation for that. Also, is there any potential connection between the upregulation of eif2a/ p-eif2a and downregulation of STAT3/p-STAT3 in GADD34-/- retina compared with the P23H RHO retina. Please add additional comments for this.
3. In figure 3C, the IL-6 expression was downregulated, while the TNFa expression was upregulated in GADD34-/- retina compared with the P23H RHO retina. Any potential explanations on these?
Author Response
Dear Reviewers,
Thank you very much for your time spent reading the manuscript and writing the comments. We also appreciate your concerns. We agree with all the comments and modified the text. We hope that the revision makes the manuscript stronger and of better quality.
Reviewer #2:
In this manuscript, the authors explored the role and potential mechanism of GADD34 in the cell apoptosis of P24H retinas. However, there are several questions still unclear and the authors should give potential explanations.
Q#1. In figure 1A and B, the ERG analysis only showed the results for the scotopic a/b wave at the light strength of 25. However, in the method part, the ERG study of retinal responses was done at several light strengths. How about those results?
Our response: Thank you. Indeed, we have more data on the ERG recording than we included in the manuscript.
Q#2. In figure 2A, there was a significant increase of eif2a and p-eif2ain GADD34-/- retina compared with the P23H RHO retina, however, the authors didn’t provide potential explanation for that. Also, is there any potential connection between the upregulation of eif2a/ p-eif2a and downregulation of STAT3/p-STAT3 in GADD34-/-retina compared with the P23H RHO retina. Please add additional comments for this.
Our response: We apologize for the confusion. The authors did not see any increase in the e-IF2a level in all three groups while p-eiF2a was indeed increased in P23H RHO and even more, increased in P23H RHO GADD34-/- retinas. This increase was the focus of the study, and we provided an explanation for it in the original submission. We also provided comments on the connection between p-eIF2a and Stat3 in the revised manuscript.
Q#3. In figure 3C, the IL-6 expression was downregulated, while the TNFa expression was upregulated in GADD34-/- retina compared with the P23H RHO retina. Any potential explanations on these?
Our response: Thank you. We included our comments in the revised manuscript.
Reviewer 3 Report (New Reviewer)
Review: “Elevation of p-eIF2a Does Not Control the Translational Rate in Degenerating Retinas with an Ongoing Integrated Stress Response”
Major:
Overall, this research study is interesting, but lacks the broader scientific impact in the vision field as it is currently presented. Up to Fig. 3, it is believed that this project is an addition to the research study published by the same group in 2019 in Cell Death and Diseases, where pEIF2a levels were studied in rd16 mice. It almost appears that both studies are supplementary with each other to this point, and in fact it would have been nice to present both together in a comparative study. An understating of the work that has gone into this research study is not attempted here and it is very much valued! It is just thought that it would be better placed in a different journal if authors prefer to present the current research study as is.
In this light: in the introduction to section 2.3, it was emphasized that in previous publications a correlation between STAT3 activation, RHO decrease and upregulation of the E3 UBR1 ligase was observed. Hence the reader was set-up for this correlation to be investigated, which is not the case. It is now questioned what changes to the E3 UBR1 ligase levels would be expected in the presence of GADD34-/- in the P23H RHO mice. It is believed that this would have been an important and essential finding that was not offered to the reader. Furthermore, the study continues with the investigation of cytokines and an inflammatory response, which is very interesting but somehow misplaced. In fact, in agreement with one of the previous reviewers, it would have been great to restructure the manuscript and make the main finding of STAT3 activation levels, inflammatory responses in conjunction with retinal degeneration in GADD24-/- P23H RHO mice a significant finding but it was missed out to highlight this appropriately. With a change of title this research study could be presented as an important correlation of sustained inhibition of protein translation, inflammation, and photoreceptor degeneration, which would have had a broader scientific impact and significance.
Lastly, in the conclusion many questions and experimental ideas were raised, that are feasible to be accomplished. It is suggested to the authors to evaluate their findings and investigate if some of their own points raised could be used to elevate their research study, such as the regulatory functions of GADD34 that ultimately lead to retinal degeneration in P23H RHO mice, or its decrease in GADD34-/- mice.
In General:
- as mentioned by one of the previous reviewers the title does NOT comprehensively describe the story line. It is thought that the research model should be stated so it is clear to the reader what research model this research study addresses.
- it is suggested to refrain from personalized statements, such as “the interest of the researchers (line 152)”, “these findings fueled the interest of the researchers further (line 175)”, “previous findings of the researchers (line 212)”, “published study of the researchers (line 224)”, “published studies of the researchers (line 231)”, “further explored by the researcher (line247)”, “it was not surprising for the researchers to learn (line 265)”, “ previous findings of the researchers (line 287)”,
Introduction:
- it is suggested to state first time abbreviations as spelled out words followed by the abbreviated word in backets, such as in line 54 “protein kinase RNA-like endoplasmic reticulum kinase (PERK)”. The reviewer asked to make this change throughout the manuscript to be consistent.
- please add appropriate references to the statement of line 59 – 62
- line 78/79 as stated: “at this time point, a few rows of photoreceptor cells are missing….” This is a statement that is anatomically incorrect! There is only one row of photoreceptor cells present int the retina. However, it is believed that there is a thinning of the ONL observed and that is what the authors indicating.
- the introduction misses an introductory section on the correlation of STAT3, IL-6 and TNFa with GADD34 and retinal degeneration.
Results:
- Figure 1: overall it appears that GADD34-/- only causes a “minor” effect of retinal degeneration in P23H RHO mice (Fig. 1A). Further, the STAT values are low and the changes in retinal thickness as presented in Fig. 1D is marginally.
- it is also questioned (as previously by other reviewers) why the C57BL6 CTR is nor present in Fig. 1A, which should be added
- please perform and present a statistical analysis for Fig. 1C
- Figure 3:
- Fig. 3B-E misses the label of LPS treatment, which makes it confusing to follow for the reader
- Fig. 3E, please add fluorescence labels
- Fig. 3D and E do not correlate with each other. It appears that Iba1+ staining is much more significant in the P23H RHO GADD34-/- mice. Hence, please perform and present a statistical analysis.
- further the data presented in Fig. 1D DO NOT correlate with presented data in 3E, where a thinning or increase in in retinal degeneration is not observable.
- in accordance with a reviewer from the first round of revisions, it is agreed that the C57BL6 mice need to be presented in all figures. The explanation by the authors in regarding this issue is not acceptable.
Discussion:
The discussion was great, raising many important and valuable point that leave the readers wanting more for the main text!
Minor:
- line 40: “200,000.0 Americans” – change to 200,000 or 200K Americans
- line 58: “… known as an integrated stress response (IRS)” – please delete “an” and correct the abbreviation for integrated stress response to ISR not IRS
- there is an inconsistency between eiF2a and eIF2a in the text, please correct
- delete a spaces in Line 62 (between study and demonstrated), Line 74 (between reported and that), line 81 (between showed and rod), line 95 (between mediated and p-eIF2a), Line 289 (between in and p-STAT3)
- line 90 – repetitive GADD34 abbreviation explanation
- line 120: “about and over two-fold” – this is contradictory, please change accordingly
- line 152/ 178: it is wrongly referred to Fig. 2B, where it should be Fig. 2D

Author Response
Dear Reviewers,
Thank you very much for your time spent reading the manuscript and writing the comments. We also appreciate your concerns. We agree with all the comments and modified the text. We hope that the revision makes the manuscript stronger and of better quality.
Review: “Elevation of p-eIF2a Does Not Control the Translational Rate in Degenerating Retinas with an Ongoing Integrated Stress Response”
Q#1. Major: Overall, this research study is interesting, but lacks the broader scientific impact in the vision field as it is currently presented. Up to Fig. 3, it is believed that this project is an addition to the research study published by the same group in 2019 in Cell Death and Diseases, where pEIF2a levels were studied in rd16 mice. It almost appears that both studies are supplementary with each other to this point, and in fact it would have been nice to present both together in a comparative study. An understating of the work that has gone into this research study is not attempted here and it is very much valued! It is just thought that it would be better placed in a different journal if authors prefer to present the current research study as is.
Our response. Thank you for mentioning our previous publication and sharing your thoughts about choosing the journals. The original manuscript was submitted for publication in a special MDPI issue on UPR. Indeed, we provided a comparison analysis between rd16 and P23H RHO mice both deficient in GADD34. We also made a comparison between the current and mentioned studies and edited the text accordingly.
In this light:
Q#2. in the introduction to section 2.3, it was emphasized that in previous publications a correlation between STAT3 activation, RHO decrease and upregulation of the E3 UBR1 ligase was observed. Hence the reader was set-up for this correlation to be investigated, which is not the case. It is now questioned what changes to the E3 UBR1 ligase levels would be expected in the presence of GADD34-/- in the P23H RHO mice. It is believed that this would have been an important and essential finding that was not offered to the reader. Furthermore, the study continues with the investigation of cytokines and an inflammatory response, which is very interesting but somehow misplaced. In fact, in agreement with one of the previous reviewers, it would have been great to restructure the manuscript and make the main finding of STAT3 activation levels, inflammatory responses in conjunction with retinal degeneration in GADD24-/- P23H RHO mice a significant finding but it was missed out to highlight this appropriately. With a change of title this research study could be presented as an important correlation of sustained inhibition of protein translation, inflammation, and photoreceptor degeneration, which would have had a broader scientific impact and significance.  Lastly, in the conclusion many questions and experimental ideas were raised, that are feasible to be accomplished. It is suggested to the authors to evaluate their findings and investigate if some of their own points raised could be used to elevate their research study, such as the regulatory functions of GADD34 that ultimately lead to retinal degeneration in P23H RHO mice, or its decrease in GADD34-/- mice.
Our response: Thank you. We carefully read the critique from other reviewers and based on the reviewer’s critiques we made a connection between p-eIF2a , Il-6, and STAT3. We also removed the text mentioning E3 UBR1ligase.
In General:
- as mentioned by one of the previous reviewers the title does NOT comprehensively describe the story line. It is thought that the research model should be stated so it is clear to the reader what research model this research study addresses.
Our response: We changed the title to “GADD34 ablation exacerbates retinal degeneration in P23H RHO mice.”
- it is suggested to refrain from personalized statements, such as “the interest of the researchers (line 152)”, “these findings fueled the interest of the researchers further (line 175)”, “previous findings of the researchers (line 212)”, “published study of the researchers (line 224)”, “published studies of the researchers (line 231)”, “further explored by the researcher (line247)”, “it was not surprising for the researchers to learn (line 265)”, “ previous findings of the researchers (line 287)”,
Our response: Thank you. We made changes accordingly.
Introduction: - it is suggested to state first time abbreviations as spelled out words followed by the abbreviated word in backets, such as in line 54 “protein kinase RNA-like endoplasmic reticulum kinase (PERK)”.
Our response: We edited the text. Thank you.
The reviewer asked to make this change throughout the manuscript to be consistent.
- please add appropriate references to the statement of line 59 – 62 - line 78/79 as stated: “at this time point, a few rows of photoreceptor cells are missing….” This is a statement that is anatomically incorrect! There is only one row of photoreceptor cells present int the retina. However, it is believed that there is a thinning of the ONL observed and that is what the authors indicating.
Our response: Thank you. We change to “row of photoreceptor nuclei”
- the introduction misses an introductory section on the correlation of STAT3, IL-6 and TNFa with GADD34 and retinal degeneration.
Our response: We moved the text related to STAT3 and GADD34 regulation from the “results” section to the introduction. Moreover, we removed the link between RHO and E3 UBR1 ligase
Results:
- Figure 1: overall it appears that GADD34-/- only causes a “minor” effect of retinal degeneration in P23H RHO mice (Fig. 1A). Further, the STAT values are low and the changes in retinal thickness as presented in Fig. 1D is marginal.
Our response: Indeed, at this point (p50), when heterozygous P23H RHO mice do not demonstrate an essential decline in ERG recording, the decline in a-wave amplitude is not dramatic although significant. Moreover, Fig. 1D does not represent the retinal thickness. It depicts the number of TUNEL-positive nuclei in the vision field.
- it is also questioned (as previously by other reviewers) why the C57BL6 CTR is nor present in Fig. 1A, which should be added - please perform and present a statistical analysis for Fig. 1C
Our response: We replaced the figure in 1A and 1C and indicated the P values in figure 1C. The C57BL6 is now provided.
- Figure 3:
- Fig. 3B-E misses the label of LPS treatment, which makes it confusing to follow for the reader
Our response: Thank you, we made changes in Figure 3 B-E.
- Fig. 3E, please add fluorescence labels
Our response: We added the fluorescence labels to the figure.
- Fig. 3D and E do not correlate with each other. It appears that Iba1+ staining is much more significant in the P23H RHO GADD34-/- mice. Hence, please perform and present a statistical analysis.
Our response: Unfortunately, the difference is not statistically significant determined by a two-way ANOVA analysis. Perhaps, if we compared individual fields using a Student t-test, the results would be significant.
- further the data presented in Fig. 1D DO NOT correlate with presented data in 3E, where a thinning or increase in in retinal degeneration is not observable.
Our response: Usually, the thickness or rows of photoreceptor nuclei is measured in the cryopreserved retinal sections stained with H&E. We did not perform this type of analysis. The two mentioned figures present the results of TUNEL and IBA staining in the retinas. We wonder what “thinning or increase in in retinal degeneration is not observable” means. Did the reviewer mean an increase in the thickness?
- in accordance with a reviewer from the first round of revisions, it is agreed that the C57BL6 mice need to be presented in all figures. The explanation by the authors in regarding this issue is not acceptable.
Our response: Thank you. We have done it.
Discussion: The discussion was great, raising many important and valuable point that leave the readers wanting more for the main text!
Minor:
- line 40: “200,000.0 Americans”
– change to 200,000 or 200K Americans
- line 58: “… known as an integrated stress response (IRS)”
– please delete “an” and correct the abbreviation for integrated stress response to ISR not IRS
- there is an inconsistency between eiF2a and eIF2a in the text, please correct
- delete a spaces in Line 62 (between study and demonstrated), Line 74 (between reported and that), line 81 (between showed and rod), line 95 (between mediated and p-eIF2a), Line 289 (between in and p-STAT3)
- line 90 – repetitive GADD34 abbreviation explanation
- line 120: “about and over two-fold” – this is contradictory, please change accordingly
- line 152/ 178: it is wrongly referred to Fig. 2B, where it should be Fig. 2D
Our response: Thank you for the careful reading. We accepted all your comments and suggested changes.
Round 2
Reviewer 2 Report (New Reviewer)
No more comments added.
Reviewer 3 Report (New Reviewer)
The reviewer would like to say thank you for taking the time to address all questions and concerns raised!
This manuscript is a resubmission of an earlier submission. The following is a list of the peer review reports and author responses from that submission.
Round 1
Reviewer 1 Report
Authors have explored a pathway related to protein synthesis in a animal model of retinal degeneration.
As regard as English language the text is well written, by the way the overall structure of the manuscript is very confusing.
- Abstract
i. there are too many abbreviations, without explanation of the acronyms.
ii. the experimental hypothesis is not clearly stated, and the aim of the study is not clear
-Introduction
This section is very short, and similarly to the abstract the experimental hypothesis was not stated. Moreover, since it appears that the animal model resembles retinitis pigmentosa the authors should reorganize their manuscript along with the title highlighting that their findings are implicated in retinitis pigmentosa.
Methods: this section is not clear. Authors must justify the animal model and compare it with other models of retinitis pigmentosa. Authors must report the number of animals per experimental group. Immunohistochemistry paragraph is very confusing, there are two different methods. The statistical analysis paragraph is embarrassing, the ANOVA stands for analysis of variance and is not aimed to multiple comparisons, the multiple comparison is carried out with post-hoc analyses.
The worse part of the manuscript is related to results and figures.
Why in some cases C57BL6J mice are reported and in other analyses not (Figure 1A)?
Figure 2. The quality of western blot is embarrassing. Protein expression normalization based on blue Coomassie is unacceptable. Why have authors not normalized the p-eIF2α and eIF2α on a given housekeeping protein?
Figure 3CDE. Authors must report the comparison with C57BL6J mice.
Figure legends. Authors report an average of N=4. Is this N related to single retinas (immunohistochemistry) or pooled retinas (western blot & PCR)? Is the N related to technical replicates or independent samples?
Reviewer 2 Report
Summary:
In the current paper the authors present their research results that were carried out in order to demonstrate that GADD34 plays an important role in the retinal degeneration process through different pathways. The subject matter is current and interesting. The authors demonstrated the multifunctional role of GADD34 in regulating the pace of retinal degeneration.
The authors presented their results in a logical and meaningful manner, using representative figures. The methods are clearly described, and the conclusions of their research are well supported by the results. Nevertheless, some minor revisions and corrections are recommended before publication.
Observations:
Line 7: Please, verify the email address and correct it if is necessary: isalt@uab.edi.
Lines 74, 110, 136, 164, 255, 259, AND 260 Please provide the meaning of the following abbreviations: “GADD34-/-”, “TUNEL”, “SUnSET”, “Iba1”, “RGC”, “INL”, AND “Prp2rds+/-”.
Line 287: Please, provide the composition of “Gonak solution”.
Lines 275, 311, 326, 351: In the description of method the authors used the “as described previously”
Lines 275, 311, 326, 351: When describing the methods, the authors usually use the phrase "as previously described" instead of describing the method. I would recommend explaining the method instead of referring to previous works.

Reviewer 3 Report
Saltykova et al. reported the effect of ablating GADD34 in P23H RHO mice. They found that GADD34 deficiency led to a reduction in scotopic ERG amplitudes and increased number of TUNEL-positive cells. I commend the authors on elucidating the UPR signaling in retinal degeneration, including RHO and rd16, which had more data. However, as outlined below, there are some concerns, which ultimately undermine my enthusiasm for this manuscript.
1. Line 110-112, although there is no statistical significant difference in RHO expression between P23H RHO GADD34−/− and P23H RHO mice, the normalized RHO in P23H RHO GADD34−/− mice was higher than P23H RHO mice. This was conflicting the ERG results in Figure 1, which showed worse response in P23H RHO GADD34−/− mice than P23H RHO mice. Anatomical histology would be helpful to show if ONL is thinner in P23H RHO GADD34−/− than P23H RHO mice.
2. When comparing the protein expression in degenerating retina, using rod-specific protein rather than beta actin might be better.
3. Figure1, what are those TUNEL positive cells?
4. It is unclear why there are more TUNEL positive cells in GADD34 P23H RHO mice.